# Active learning regression quality prediction model and grinding mechanism for ceramic bearing grinding processing

Longfei Gao[1,2]*, Yuhou Wu[2], Jian Sun[2], Junxing Tian[2]

1 School of Engineering Training and Innovation, Shenyang Jianzhu University, Shenyang, China,
2 School of Mechanical Engineering, Shenyang Jianzhu University, Shenyang, China

* sjian2024@126.com

## Abstract

The study aims to explore quality prediction in ceramic bearing grinding processing, with particular focus on the effect of grinding parameters on surface roughness. The study uses active learning regression model for model construction and optimization, and empirical analysis of surface quality under different grinding conditions. At the same time, various deep learning models are utilized to conduct experiments on quality prediction in grinding processing. The experimental setup covers a variety of grinding parameters, including grinding wheel linear speed, grinding depth and feed rate, to ensure the accuracy and reliability of the model under different conditions. According to the experimental results, when the grinding depth increases to 21 µm, the average training loss of the model further decreases to 0.03622, and the surface roughness Ra value significantly decreases to 0.1624 µm. In addition, the experiment also found that increasing the grinding wheel linear velocity and moderately adjusting the grinding depth can significantly improve the machining quality. For example, when the grinding wheel linear velocity is 45 m/s and the grinding depth is 0.015 mm, the Ra value drops to 0.1876 µm. The results of the study not only provide theoretical support for the grinding processing of ceramic bearings, but also provide a basis for the optimization of grinding parameters in actual production, which has an important industrial application value.

## 1. Introduction

With the increasing demand for high-performance bearings in modern industry, ceramic materials are gradually becoming an important choice in the field of bearing manufacturing due to their excellent wear resistance and rigidity [1]. In particular, L. D. M. Fernandes et al. discovered that ceramics exhibit excellent mechanical properties under high - temperature, high - speed, and high - load conditions. This makes them ideal for bearing applications, yet also poses unique challenges during grinding [2]. However, U. Patel and V. Patel pointed out that traditional grinding processing (GP) often fails to meet the surface roughness (SR) requirements when grinding high - hardness ceramic bearings. This not only leads to subpar bearing performance but also highlights the significant difference between grinding ceramic

**Data availability statement:** All relevant data are within the manuscript and its Supporting Information files.

**Funding:** The research is supported by National Natural Science Foundation of China in 2024: Research on Dynamic Characteristics Control of Full Ceramic Ball Bearing Retainers for Wide Temperature Range Oil free Lubrication Conditions (Fund No. 52405123); National Natural Science Foundation of China in 2021: Research on the Characteristics and Discrimination Mechanism of Lubricating Oil Film in Multi Field Coupled All Ceramic Ball Bearings Based on Elastic Flow Pressure Lubrication (Fund No. 52105196). The funders had no role in study design, data collection and analysis, decision to publish, or preparation of the manuscript.

**Competing interests:** The authors have declared that no competing interests exist.

bearings and other materials. Ceramic bearings demand higher precision in surface finish due to their application in high - performance equipment, and traditional GP struggles to deliver [3]. Electrolytic in - process dressing (ELID) grinding shows promise for optimizing grinding quality and efficiency. But as D. Xiang et al. and B. Dou et al. revealed, most existing studies mainly focus on single grinding parameters. They lack a comprehensive exploration of the multilevel effects of material properties and grinding conditions specific to ceramic bearings. For instance, the unique hardness and brittleness of ceramic materials during bearing grinding can result in different chip formation and heat dissipation patterns compared to other materials. These differences require a more in - depth understanding of the grinding mechanism and better - tailored quality prediction models [4–5]. The process of grinding ceramic bearings is significantly different from grinding materials such as metals or plastics. Ceramic materials typically have extremely high hardness and low toughness, which makes them prone to brittle fracture and surface cracks during grinding. In contrast, metal materials are more prone to plastic deformation during grinding, and their stress distribution is relatively uniform. In addition, the thermal sensitivity of ceramic materials makes them more sensitive to the heat generated during grinding, and excessively high temperatures may lead to a decrease in material properties and deterioration of surface quality. Therefore, the study of grinding parameters and methods for ceramic bearings is particularly crucial to prevent the generation of cracks and ensure higher machining quality. In past studies, some scholars have explored the effects of multiple grinding wheels (GWs) on SR, such as analyzing the performance of diamond and CBN wheels in different grinding stages. However, these studies still have shortcomings. First, it lacks an in-depth understanding of the physical mechanism of the GP, making it difficult to systematically regulate the grinding parameters. Second, thorough effect studies on various material combinations are lacking. Conclusions are often drawn under specific conditions, and there is a lack of broad applicability analysis for a wide range of operating conditions and material properties. Third, the current research fails to fully utilize modern data analysis techniques such as machine learning, and there is a large untapped potential in the utilization of unlabeled data [6–7].

The grinding mechanism is very important for ceramic bearing GPing. S. M. Lee et al. investigated the effects of material properties and grinding conditions on the SR of G5 grade bearing steel balls ground with ELID on ceramic materials. The feasibility was verified experimentally using high efficiency ELID grinding. The outcomes revealed that the grinding SR of #2000 diamond GW could reach 0.014 μm, which meets the requirements. The #325 CBN wheel performed well in rough grinding. Overall, it indicated that these two abrasive combinations were suitable for high-precision machining of silicon nitride ceramics [8]. L. Jiang et al. designed a new grinding head to improve the grinding efficiency and quality of high hardness ceramic composite Cf/SiC. The study defined the "effective area ratio" as the design parameter. Experiments indicated that the new grinding head could significantly reduce the grinding force and SR by 73% and 10%, respectively, and effectively improve the grinding quality [9]. X. Wang et al. systematically analyzed surface finishing techniques for rolling elements of precision bearings, including centerless grinding and superfinishing. The study compared SR with roundness error, thus predicting future trends in the technology, emphasizing its importance for bearing performance and reliability [10]. A review of the use of advanced ceramics and coatings in fuel and mineral processing was conducted by E. Medvedovski et al. The study analyzed material selection, manufacturing processes and their examples in erosive wear, emphasizing the need for high reliability and complex shaped materials with coating optimized processing solutions [11]. X Chen et al. analyzed the friction characteristics of a hybrid ceramic bearing of silicon carbide and GCr15 steel under non-lubricated and water-lubricated conditions. The study

compared the frictional properties of texturized and non-texturized SiC surfaces under the two conditions. The results indicated that its friction coefficient increased in the dry friction condition and decreased in the water-lubricated condition. Laser texturization could help to improve the frictional properties of SiC [12].

S Mahmood et al. developed a machine learning based energy consumption prediction model for the problem of green building energy consumption exceeding design expectations. The study used active learning and algorithms such as random forest and decision tree to predict the cooling and heating energy consumption of green buildings and processed the data through data visualization and Z-Score normalization. The outcomes indicated that the model had high accuracy in cooling and heating energy consumption prediction. This could effectively reduce energy consumption, improve building sustainability, and provide a new method for green building energy management [13]. F. M. Talaat et al. proposed an algorithm for crop yield prediction with IoT technology. The study combined multiple data models for prediction and developed an extreme tree regression model that showed the best performance to support agricultural decision making [14]. F Feyzi et al. proposed two deep learning based defect prediction models to address data complexity and category imbalance in software defect prediction. Moreover, static code metrics with pool-based active learning strategy, as well as nearest neighbor and KNN algorithms were used to deal with category imbalance. Experimental results on GitHub Bug Dataset and Unified Bug Dataset indicated that the proposed models improved 13% to 14% and 11% in F-measure and AUC metrics, respectively, which was compared with traditional machine learning algorithms. It verified the effectiveness of the models [15]. Predicting the material removal (MR) rate during chemomechanical leveling was done using the semi-supervised deep kernel active learning model that C. Lv et al. proposed. The findings demonstrated that, for various labeled sample ratios, the model's prediction accuracy (PA) outperformed that of the conventional approach [16]. L. Zhu et al. proposed an economical data-driven power theft detection method. This study reduced the data labeling cost through deep active learning and verified its reliability and effectiveness in power theft monitoring [17].

In summary, many experts have conducted research on high-precision GPes and machine learning-based energy consumption prediction. However, the current research still suffers from the lack of in-depth understanding of the grinding mechanism, the inadequacy of achieving synergistic optimization of multiple materials, and the failure to adequately integrate data-driven models with traditional machining techniques. Therefore, the study proposes an innovative active learning-based regression quality prediction model. The model is used to dynamically predict and optimize material properties and process parameters during the GP. Systematic experiments are conducted to observe the effect of different grinding conditions on ceramic SR. Data-driven machine learning techniques are further combined to improve the understanding and control of the GP. The study proposes the following hypothesis. Firstly, by combining active learning and deep learning models, the accuracy and reliability of surface quality prediction in ceramic bearing grinding can be significantly improved. Then, in the grinding process of ceramic bearings, optimizing grinding parameters (including wheel linear velocity, grinding depth, and feed rate) can significantly improve surface roughness, and the relative error range of the prediction model can be controlled within a small range. Finally, the optimization process based on active learning regression quality prediction model can effectively reduce the cost of experimental data annotation and improve the efficiency of grinding parameter optimization. It is expected to contribute to the quality improvement and energy management of ceramic bearing GP, and to contribute new ideas and methods for the efficient and high-precision processing of ceramic bearings in the future.

## 2. Methods and materials

### 2.1. Grinding mechanism analysis and material removal mechanism

Grinding is a high-precision machining technique that relies on the rapid rotation of an abrasive wheel to cut a workpiece. The abrasive grains (AGs) on the GW remove material by scraping, plowing, cutting to obtain precise dimensions and surface quality [18]. Three steps make up the GP: The grinding depth (GD) is shallow and the GW grits are in touch with the workpiece during the first contact stage. The workpiece material undergoes mainly elastic deformation and no chips are formed. In the material deformation stage, the embedding depth of the AG increases. Elastic and plastic deformation of the workpiece surface material occurs, forming grooves and bulges. In the MR stage, the embedding depth of AGs exceeds a critical value. The material forms abrasive chips under shear and detaches from the workpiece surface, leaving deeper grooves, bulges and scratches [19]. The three stages of abrasive cutting process are shown in Fig 1.

According to the analysis of the interaction between the AG and the workpiece, the MR in the GP is mainly realized in the chip-forming stage, while the sliding rubbing and plowing stage is the preparation stage. To improve the MR efficiency, the proportion of the chip-forming stage in the overall GP should be increased. To encourage more grits to reach the chip formation stage, this can be accomplished by modifying the grinding settings, such as deepening the grits' embedding or enhancing the GW's grit distribution [20]. At the same time, grinding forces and temperatures need to be monitored to ensure process stability and workpiece surface quality.

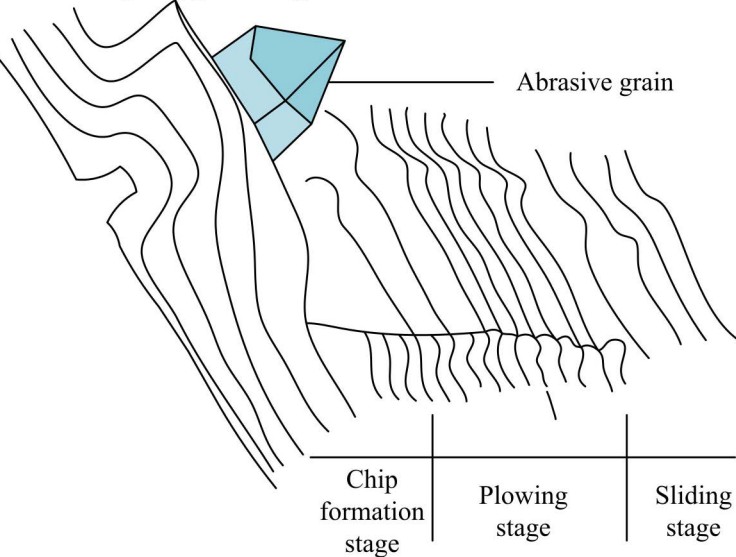

Chip formation stage:Abrasive particles cut materials to form chips.
Plowing stage: The abrasive particles form grooves under the surface of the material, causing deformation of the material.
Sliding stage: Abrasive particles slide out from the surface of the workpiece, generating frictional heat.

**Fig 1. Three stages of abrasive cutting process.**

The indentation fracture mechanics model simplifies the contact between a single abrasive grit and a hard brittle material to a radial force exerted on the material by a regular indenter in order to simulate the MR mechanism during the GP [21]. The tiny action between the GW and the workpiece is found to be comparable to indentation tests when grinding hard and brittle materials, and the abrasive grit action can be thought of as a little indentation operation. Although the AGs on the GW have different shapes, they are usually abstracted as Vickers tetrahedral model in the study, as shown in Fig 2.

The deformation and crack formation process induced by the Vickers indenter on the ceramic surface under quasi-static loading conditions. At low initial loads, only localized plastic deformation occurs below the indenter. When the load exceeds the critical value P1, radial cracks appear under the indenter and expand with increasing load. When the critical value P2 is further exceeded, transverse cracks form and expand, eventually leading to MR. The expression of the critical load Pm that produces radial cracks is shown in Equation (1).

$$P_{\mathrm{m}} = 54.5(\alpha/\eta^2\gamma^2)(\frac{K_{IC}^4}{H^3}) \tag{1}$$

In Equation (1), $\alpha$ represents constants related to material properties. $\eta$ and $\gamma$ are related to the geometry of the material or specific conditions during the GP. $K_{IC}$ is the fracture toughness of the material. $H$ is the hardness of the material. The expression for the critical load Pl that produces radial cracks is shown in Equation (2).

$$P_L = \zeta(\frac{K_{IC}^4}{H^3})f(\frac{E}{H}) \tag{2}$$

In Equation (2), $\zeta$ is a dimensionless coefficient. $f(\frac{E}{H})$ represents the mechanical behavior of the material and the complex interactions during the GP. It depends on the ratio of elastic modulus $E$ to hardness $H$. Both formulas are used to calculate the critical value for radial crack generation, but they consider different factors and focus. Formula 1 is more directly related to the fracture toughness and hardness of materials, while Formula 2 further considers the ratio of elastic modulus to hardness, which has a more complex impact on the formation of cracks in materials. The generation of cracks has a negative impact on the material properties, therefore, suitable process parameters should be selected to minimize crack generation in practical processing. During the GP of ceramic materials, once the force exerted on the AG exceeds a specific threshold, cracks of length Cl are initiated on the surface of the workpiece, as shown in Fig 3.

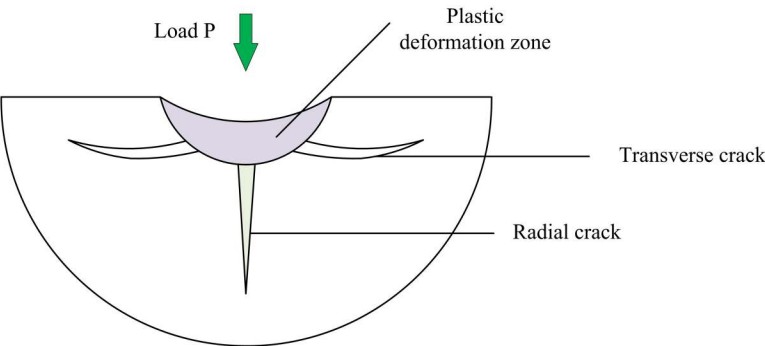

**Fig 2. Indentation fracture mechanics model.**

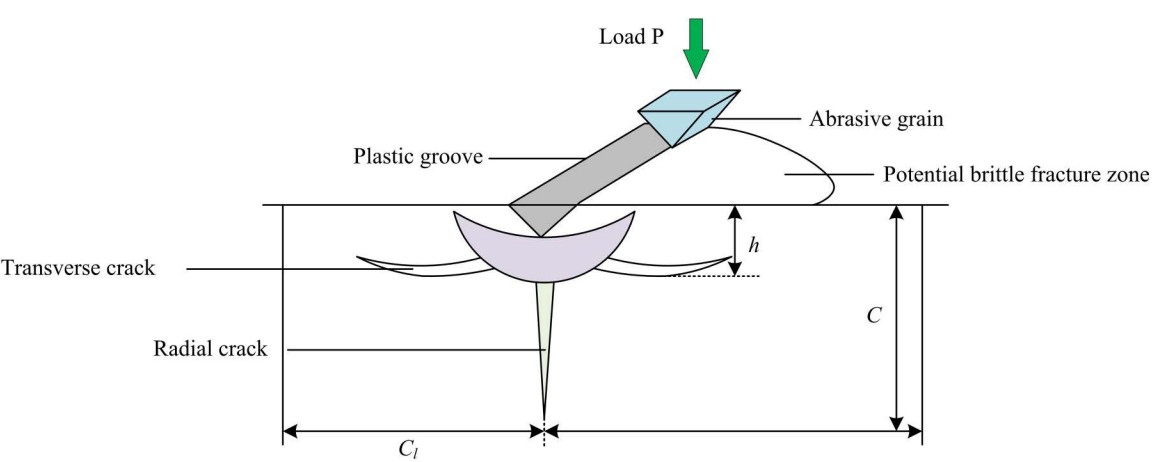

**Fig 3. Grinding particle cutting machining model.**

As the force increases, the cracks expand, triggering the formation and expansion of microcracks on the surface, which ultimately leads to spalling and cratering of the material, completing the removal of the material [22]. Based on the GP model, a surface damage model is constructed. The model accurately evaluates the damage rate of the ground surface by measuring the crater area and the number of AGs involved in grinding. The crack length is shown in Equation (3).

$$C_l = C^l \left[ 1 - \left( F_0 / F_{n'} \right)^{1/4} \right]^{1/2} \tag{3}$$

In Equation (3), $C^l$ is a reference value for crack length. $F_0$ represents the initial load applied to the material. $F_{n'}$ represents the load normal to a single AG of the material.

$$F_0 \equiv (\zeta_a / A_0^2)(\cot\psi)^{-2/3}(K_c^4 / H^3)(E / H) \tag{4}$$

In Equation (4), $\zeta_a$ represents the correlation coefficient of the GW. $A_0$ represents the initial area. $\cot\psi$ is the inverse of the cotangent function. $K_c$ represents the ability of the material to resist crack extension in the presence of cracks. $H$ denotes the ability of the material to resist plastic deformation. $E$ denotes the modulus of elasticity of the material.

$$C^l \equiv \left[ \zeta_b \left( \cot\psi \right)^{5/6} A_0^{-1/2} \left( K_C H \right)^{-1} E^{3/4} \right]^{1/2} F_n^{'5/8} \tag{5}$$

In Equation (5), $\zeta_b$ denotes the correlation coefficient of the GW. When $F_n^{'} \gg F_0$, Equation (6) is obtained.

$$C_l \equiv \left[ \zeta_b \left( \cot\psi \right)^{5/6} A_0^{-1/2} \left( K_C H \right)^{-1} E^{3/4} \right]^{1/2} F_n^{'5/8} \tag{6}$$

The whole equation of Equation (6) combines the geometrical properties, mechanical properties and external loading conditions of the material in order to calculate the crack length. The expressions for the two quantities $D_{sA}$ and $N_s$ related to crack formation and extension during material grinding are shown in Equation (7).

$$\begin{cases} D_{sA} = \pi C_l^2 N_s A / A = \pi C_l^2 N_s \\ N_s = 4f / \left[ d_g^2 \left( 4\pi / 3b \right)^{2/3} \right] \end{cases} \quad (7)$$

In Equation (7), $D_{sA}$ represents the surface area broken with the grinding surface. $N_s$ represents a dimensionless factor. $f$ represents the proportion of abrasive actually involved in GP. $d_g$ represents the grain size of the GW. $b$ denotes the grinding cross diameter. The system of equations relates the crack length $C_l$ to the parameters of the GP, allowing the calculation of the surface area or volume metric associated with the crack [23]. The grinding technique is used for precision machining of external surfaces of rotating body parts such as cylindrical, conical and multi-step shafts, capable of achieving accuracy of IT5 to IT8 and SR of 0.8 to 0.2 µm [24]. Four main grinding methods, longitudinal, plunge, depth, and hybrid, are used when performing external grinding [25]. Fig 4 displays the geometric contact arc length model.

In Fig 4, grinding is the cumulative effect of numerous AGs on a GW making tiny cuts in a workpiece. The grinding temperature and surface quality are significantly impacted by the contact arc length. In the geometric model, the workpiece and the GW are treated as rigid bodies. The effect of elastic deformation on the contact arc length is not taken into account and the relationship between the contact arc length of the GW and the workpiece is derived from the geometric analysis. The geometric parameters in GP are calculated as shown in Equation (8).

$$l_g = \sqrt{a_p \cdot d_e} \quad (8)$$

In Equation (8), $l_g$ denotes the length of the grinding path. $a_p$ denotes the transverse feed in the GP. $d_e$ denotes the equivalent diameter of the GW. The MR rate is a key index for evaluating the rate of MR during the GP, which reflects the volume of material removed per unit time, as shown in Equation (9).

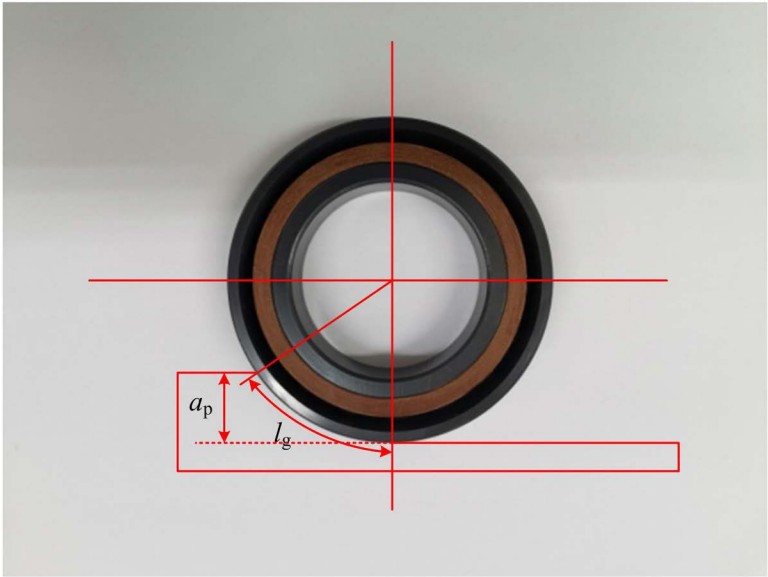

**Fig 4. Geometric contact arc length model.**

$$Q_w' = \nu_w \cdot a_p \tag{9}$$

In Equation (9), $Q_w'$ denotes the MR rate. $\nu_w$ denotes the rate of surface movement of the GW. $a_p$ denotes the depth of grinding. GP is actually the cumulative effect of MR from a material surface by numerous single AGs. According to the theory of equivalent chip thickness of single grit grinding, MR can be interpreted as the grits cutting the material continuously with the linear velocity of the GW, as shown in Equation (10).

$$h_{eq} = \frac{\nu_w}{\nu_s} a_p \tag{10}$$

In Equation (10), $h_{eq}$ denotes the thickness of material removed in a unit of time, determined by the combination of wheel linear velocity and GD. $\nu_s$ denotes the rate of workpiece surface movement. The formula can be employed to predict the MR efficiency of the GP under specific grinding parameters, so as to optimize the GP.

## 2.2. CNN-LSTM-based quality prediction model and active learning strategy for ceramic bearing grinding

The accuracy and surface quality of GP is the result of the coordination of multiple process parameters. Only through the adjustment of a single parameter, it is usually impossible to fully understand the machining process, which leads to the failure of machining quality control [26]. Meanwhile, the extraction of time-domain and spatial information of the grinding parameter sequences also affects the effectiveness of quality prediction. The parameter variability of various grinding operations and the parameter changes of the same activities due to equipment and material variables must also be taken into consideration in large-scale production datasets [27–28]. To optimize the above problems, the study designed a ceramic bearing grinding quality prediction model based on convolutional neural network (CNN) and long short-term memory (LSTM) in the context of local process parameter characteristics. It is also combined with active learning model to improve the accuracy and efficiency of prediction [29–30]. The CNN model is shown in Fig 5.

The convolution process is affected by parameters such as convolution kernel size (CKS) and step size. The output size $m \times m$ of the convolution is shown in Equation (11).

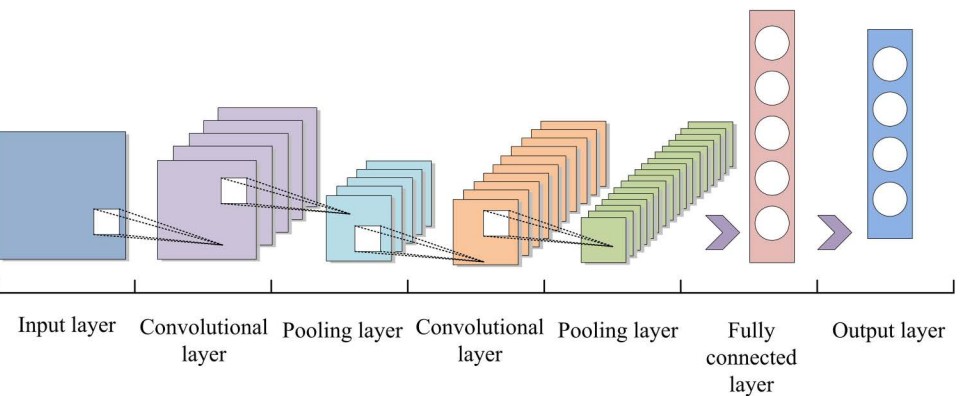

**Fig 5. CNN model.**

$$m = \frac{(N+2p)-k+1}{s} \tag{11}$$

In Equation (11), $N \times N$ denotes the input size. $k \times k$ is the CKS. $s$ is the step size. $p$ denotes the number of zero padding.      is the downward rounding. The convolution is followed by downsampling through the pooling layer for feature fusion and smoothing. The convolution output $y_{d,i^l,j^l}$ at the corresponding position is shown in Equation (12).

$$y_{d,i^l,j^l} = \sigma\left(\sum_{c^l=0}^{C^l}\sum_{i=0}^{h^l}\sum_{j=0}^{w^l} p'_{d,c^l,i,j} \times x_{c^l,i^l+i,j^l+j} + b_d\right) \tag{12}$$

In Equation (12), $C^l$ denotes the input channels' quantity in layer $l$. $C^l \times h^l \times w^l$ denotes the individual convolutional kernel size. $d$ denotes neuron number. $i^l$ / $j^l$ are positional information. $b$ denotes the corresponding bias. $p'$ denotes convolutional kernel parameters. $\sigma$ denotes the activation function. Among them, the constraints of position information are shown in Equation (13).

$$\begin{cases} 0 \le i^l \le H^l - h^l + 1 \\ 0 \le j^l \le W^l - w^l + 1 \end{cases} \tag{13}$$

In Equation (13), $C^l \times H^l \times W^l$ denotes the tensor of layer $l$ input. In ceramic bearing GP quality prediction, due to the time-dependent nature of the GP, traditional CNNs cannot effectively deal with such problems. LSTM, as an improved version of RNN, solves the gradient vanishing and exploding problems through gating mechanism and is able to capture long-term dependencies. This makes LSTM suitable for time-series analysis of parameters in the GP, helping the model to maintain the memory of historical information and thus accurately predict the grinding quality. The LSTM model is shown in Fig 6.

The input gate $i$ is calculated as shown in Equation (14).

$$i_t = \sigma\left(W_i \bullet [h_{t-1}, x_t] + b_i\right) \tag{14}$$

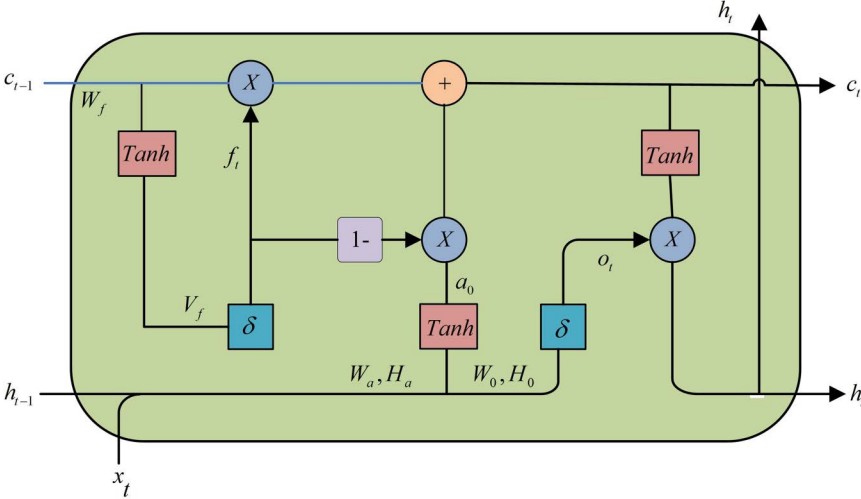

**Fig 6. LSTM model.**

In Equation (14), $W$ is the learning weight of the corresponding gate. $h_{t-1}$ is the hidden state of the previous moment. $x_t$ is the input vector of the moment. $\sigma(\cdot)$ denotes the activation function. b denotes the corresponding learning bias. The computation of the oblivious gate $f$ is shown in Equation (15).

$$f_t = \sigma\left(W_f \bullet [h_{t-1}, x_t] + b_f\right) \tag{15}$$

The output gate $o$ is calculated as shown in Equation (16).

$$o_t = \sigma\left(W_o \bullet [h_{t-1}, x_t] + b_o\right) \tag{16}$$

The data preparation before the network operation has been designed in the previous section, while the role of the whole composite neural network is to extract and classify the motion features. The overall flow is shown in Fig 7.

In Fig 7, in the field of ceramic bearing GP, deep localized features in the GP can be deeply extracted by setting up a multi-stream network with multiple streams. Moreover, the weights of each multi-stream network are different to adapt to different grinding conditions and parameters. Before the local feature extraction, the extraction of the underlying grinding parameter features is first performed by a layer of shared LSTM-CNN. This helps to reduce the computational burden of the model and speeds up the convergence of the model. With the connection of multiple LSTM-CNN layers, the network is able to cross-extract spatio-temporal data to ensure the preservation of domain information during the iteration process and realize the transformation of advanced semantic features. Multi-stream networks contain multiple tributaries with different structural parameters. These tributaries can work in parallel to realize the extraction of different features, and their gradient propagation is relatively independent. Eventually, the accuracy of predicting grinding quality is increased by fusing the data taken from various streams to create multimodal grinding information. The active learning model screens representative samples in the branched streams. This optimizes the learning effect and improves the PA through feature fusion. In grinding quality prediction, active learning reduces the dependence on a large amount of labeled data by selecting the most informative samples for labeling, reduces the cost and improves the learning efficiency. The uncertainty measure of the computed samples is shown in Equation (17).

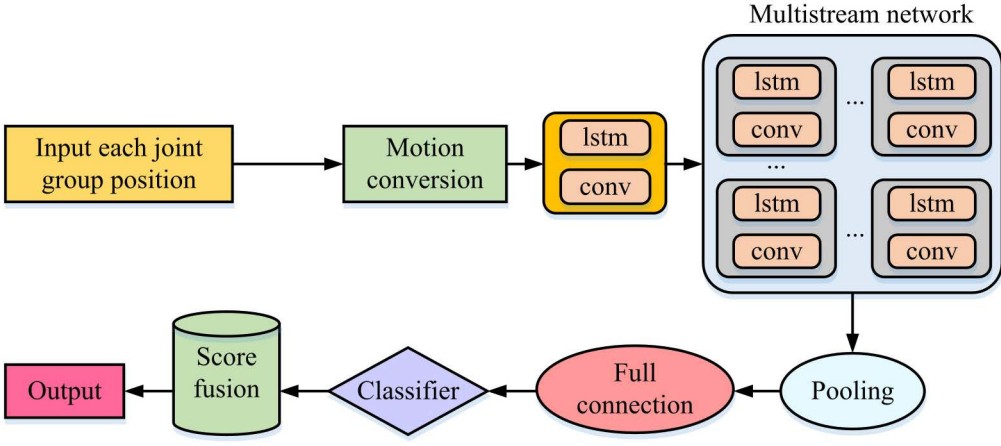

**Fig 7. CNN-LSTM action recognition process.**

$$U(x) = 1 - max(p(y|x)) \tag{17}$$

In Equation (17), $U(x)$ represents the uncertainty of sample $x$. $(p(y|x))$ is the predicted probability distribution of the model for sample $x$. The samples to be labeled in the next step can be selected based on the value of $U(x)$. The model weights are updated as shown in Equation (18).

$$w_{t+1} = w_t - \eta \nabla L(w_t, D_t) \tag{18}$$

In Equation (18), $w_t$ is the weight of the current model. $\eta$ is the learning rate (LR). $L$ is the loss function. $D_t$ is the current labeled data set. Then the next data point is selected as shown in Equation (19).

$$x^* = \arg\max_{x_i \in D'} U(x_i, w_t) \tag{19}$$

In Equation (19), $D'$ is the unlabeled data set. $D'$ is the uncertainty measure for the unlabeled data point $x_i$. Then, by continuously updating the weights and choosing the samples that best enhance the model's performance, effective learning is accomplished. The study names the proposed model as AL-CLSTM.

## 3. Results

### 3.1. Model performance comparison and optimization evaluation

The grinding machine used in the experiment is the YE7232 model from Qinchuan Machine Tool Group Co., Ltd. It is equipped with a 400mm diameter grinding wheel made of cubic boron nitride (CBN) with a particle size of 100 #. We chose this type of grinding wheel because it exhibits excellent grinding efficiency and surface quality when grinding ceramic materials. In addition, the measurement of surface roughness was carried out using the JB-1C model measuring equipment from Shanghai Precision Instrument Co., Ltd., which has high precision and repeatability, ensuring the reliability and accuracy of the data.

For verifying the optimization effect of the research-designed module on the overall algorithm, the study selects three models to compare with the research-designed model. The study first conducted experimental analysis on the performance of the algorithm itself. Based on commonly used hyperparameter settings in relevant literature, key parameters such as learning rate, training epochs, and standardization parameters were preliminarily selected. Subsequently, these parameters were optimized using grid search methods to determine the optimal combination, ensuring the convergence of model training and the accuracy of predictions. In addition, hardware configuration is also selected based on experimental computing requirements to support efficient processing of large-scale data[31]. The experimental environment and related algorithm parameter settings are shown in Table 1.

In Table 1, the experimental environment configuration includes Ubuntu operating system with NVIDIA GeForce RTX 3080 graphics card. The machine learning framework used is PyTorch 1.10.0, and the programming language is Python 3.8. Model training is performed for a total of 100 rounds, and the initial LR is set to 0.001. The mean value of data normalization is 0.02, and the bias parameter is 0. The initial weight of score fusion is 0.5. The step size is set to 1 during optimization, and the activation function is selected as ReLU. The model training loss values under different LRs are shown in Fig 8.

**Table 1.  Experimental environment and related algorithm parameter settings.**

| Name | Environment/ParametersGPU |
|------|---------------------------|
| Experimental system | Ubuntu |
| GPU | RTX 3080 |
| Machine learning framework | Pytorch |
| Programming language | Python |
| Training rounds | 80 |
| Initial learning rate | 0.001 |
| Standardized mean | 0.02 |
| Bias parameter | 0 |
| Initial weight of score fusion | 0.5 |
| Step size | 1 |
| Activation function | ReLU |

Fig 8(a) demonstrates the variation of loss values during model training for a fixed LR of 0.06. The loss value gradually drops from a larger initial value as the repetitions increases. The horizontal axis (HA) of the graph denotes the iterations and the vertical axis (VA) denotes the model training loss value. Within 2000 iterations, the loss value decreases from around 0.80 to a range of 0.02 to 0.04. Fig 8(b) illustrates the trend of the model training loss value when LR is 0.006. HA is still the number of iterations, and VA represents the loss value. The loss value starts at approximately 0.07 and gradually decreases to around 0.001 as the number of iterations increases. Fig 8(c) shows the variation of model training loss values with the number of iterations when the LR is reduced to 0.0006. The HA and VA denote the iterations and the loss value. The loss value starts at 0.80 and gradually decreases to a range of 0.00 to 0.10 as the iterations proceed. Fig 8(d) illustrates the case where the LR is 0.00006. change in the model training loss value. The horizontal and vertical axes have the same meaning as in the previous three figures. The loss value starts at 0.80 and gradually decreases to around 0.001 as the iterations increases. When the learning rate is too low, the step size of parameter updates becomes smaller, causing the model to wander around the optimal solution of the loss function, and the convergence process appears very slow. The comparison of the performance curves of the two models is shown in Fig 9.

Fig 9(a) displays the ROC curves of CNN-LSTM and AL-CLSTM models. At a false positive rate of 0.4%, the AL-CLSTM model achieves a sensitivity of 92%. This indicates that at a lower false positive rate, AL-CLSTM can effectively recognize positive class samples, demonstrating its advantage in recognition ability. Overall, the AL-CLSTM curve is higher than CNN-LSTM in the entire range of false positive rates, indicating that AL-CLSTM has better performance. Fig 9(b) presents the performance of AL-CLSTM and CNN-LSTM models in terms of precision and recall. The precision of AL-CLSTM is about 80% when the recall rate reaches 80%. This performance is outstanding compared to the CNN-LSTM model. This indicates that AL-CLSTM can effectively reduce false positive samples while maintaining a high recall rate. In addition, the PR curve of AL-CLSTM is relatively smooth, indicating that the model is more adaptable to different thresholds. In Fig 9(c), AL-CLSTM has a larger correctly classified region and a smaller misclassified region under different thresholds, reflecting its superior performance, and the threshold adjustment can further optimize the classification results and enhance the reliability of the model prediction. This suggests that by reasonably setting the threshold value, not only can the correct classification ability of the model be improved, but also can effectively reduce the misclassification, thus optimizing the overall performance of the model.

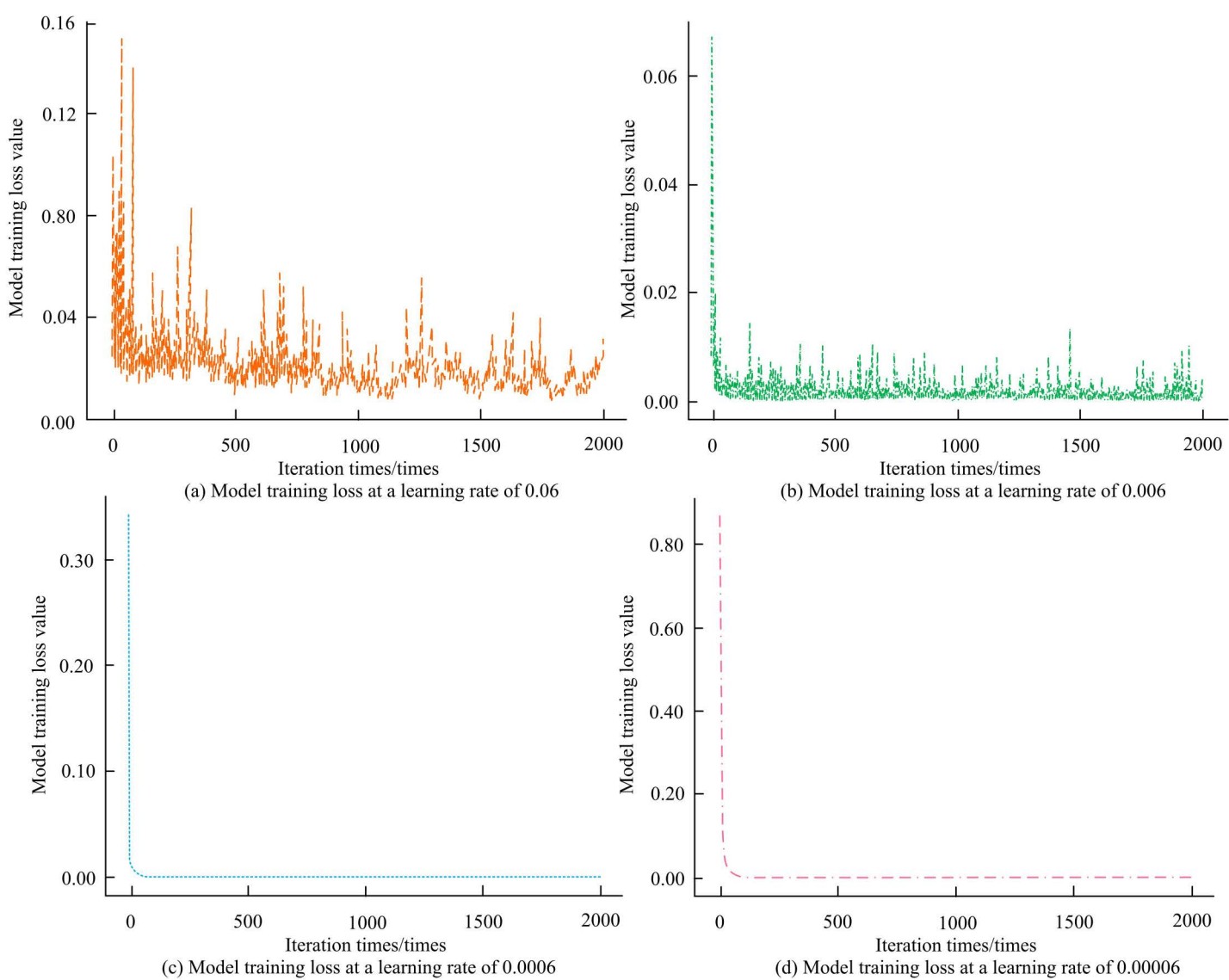

**Fig 8. Model training loss values at different learning rates.**

Fig 10(a) demonstrates the relationship between the number of iterations and the error value for different models used in the CWRU bearing data center. The error value of the AL-CLSTM model decreases to nearly 0.2 when the number of iterations reaches 70, showing high efficiency and stability. In contrast, the error values of the DTRL and CNN-SBLSTM models are 1.2 and 0.3 at the same iteration step, showing a clear performance gap. In addition, the error of the CNN-LSTM model is higher at the beginning of the iteration number, but gradually decreases to about 0.1, showing good convergence. Fig 10(b) shows the iterations versus error values for different models at the Paderborn University data center. The error value of AL-CLSTM is about 0.1 at 70 iterations, which remains low and is similar to the performance of the CWRU data center. However, the CNN-LSTM model slightly outperforms the CWRU data center here, with the final error dropping to 0.05. The DTRL model has an error higher than 1.2, showing a slight lack of adaptability to the data.

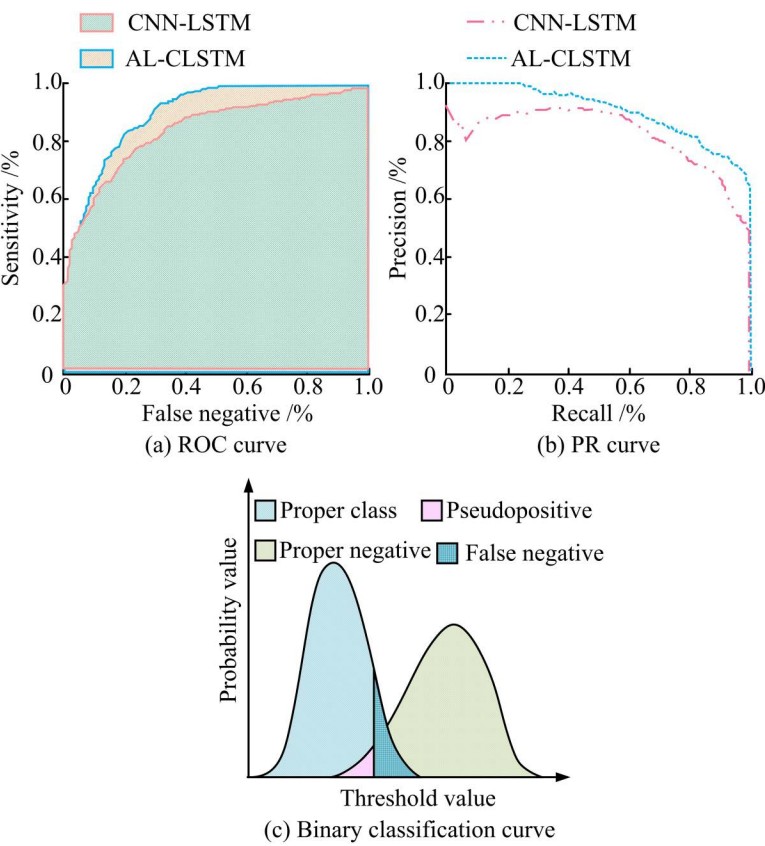

(a) ROC curve  (b) PR curve

(c) Binary classification curve

**Fig 9. Comparison of the performance curves of the two models.**

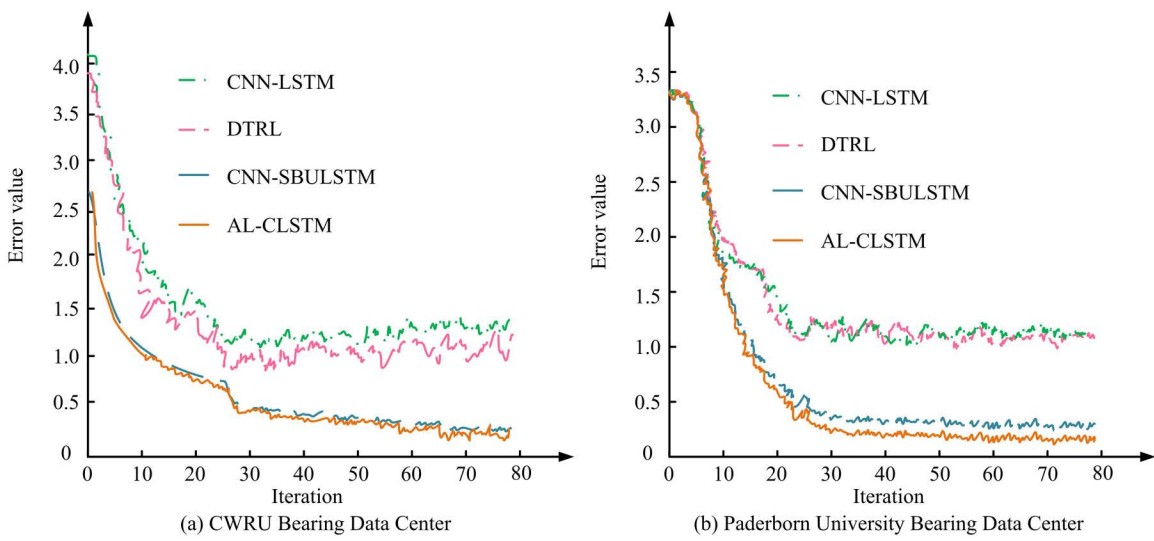

(a) CWRU Bearing Data Center  (b) Paderborn University Bearing Data Center

**Fig 10. Comparison of model errors in different volume datasets.**

## 3.2. Experimental evaluation and result analysis of ceramic bearing grinding processing

Experiments are conducted to evaluate the active learning regression quality prediction capability of different models in ceramic bearing GP. A variety of deep learning models are used in the experimental setup, including AL-CLSTM, CNN-LSTM, CNN-SBLSTM, and DTRL. All the models are trained under uniform grinding parameters, and the selected LR, number of iterations, and evaluation metrics are adjusted by several experiments to ensure the reliability of the data. The experimental results of active learning regression quality prediction model (AL-CLSTM) for ceramic bearing GP are shown in Table 2.

In Table 2, the AL-CLSTM model has the lowest average training loss of 0.03572 after a LR of 0.06 and 2000 iterations. Moreover, the average prediction error is only 0.04589, showing excellent prediction performance. Its convergence time is 250 s, and its accuracy reaches 94.32%. In contrast, the DTRL model performs poorly under the same number of iterations, with an average training loss of 0.06509, an average prediction error of 0.08072, and an accuracy of only 86.90%. This result indicates that the active learning approach significantly outperforms the traditional regression model in ceramic bearing GP quality prediction and warrants further investigation. When AL-CLSTM has 3000 iterations and a learning rate of 0.03, it still exhibits better average training loss and prediction error. This indicates that even with a decrease in learning rate, the model can still maintain good convergence and accuracy. The experimental results of AL-CLSTM for ceramic bearing GP are shown in Table 3.

Table 3 summarizes the experimental results of AL-CLSTM for ceramic bearing GP under different parameter settings. When the grinding depth is 20 μ m, the average training loss of the AL-CLSTM model is 0.04237, and the average prediction error is 0.05312, indicating that this setting can basically achieve good prediction performance. However, when the grinding depth increased to 21 μ m, the average training loss of the model further decreased to 0.03622, and the surface roughness Ra value also significantly decreased

**Table 2. Experimental results of AL-CLSTM for ceramic bearing grinding processing.**

| Model | Iteration count | Learning rate | Average training loss | Average prediction error | Convergence time (s) | Accuracy (%) |
|---|---|---|---|---|---|---|
| AL-CLSTM | 2000 | 0.06 | 0.03572 | 0.04589 | 250 | 94.32 |
| CNN-LSTM | 2000 | 0.006 | 0.02714 | 0.03854 | 280 | 92.17 |
| CNN-SBLSTM | 2000 | 0.0006 | 0.04122 | 0.05211 | 320 | 90.45 |
| DTRL | 2000 | 0.00006 | 0.06509 | 0.08072 | 450 | 86.90 |
| Baseline model (linear regression) | 2000 | N/A | 0.08945 | 0.10234 | 180 | 82.50 |
| CNN-RNN | 2500 | 0.002 | 0.03418 | 0.04132 | 350 | 91.80 |
| SAX-Predict | 1500 | 0.004 | 0.04734 | 0.05648 | 310 | 88.25 |

**Table 3. Experimental results of AL-CLSTM for ceramic bearing grinding processing.**

| Model | Parameter settings | Iteration count | Learning rate | Average training loss | Average prediction error | Convergence time (s) | Surface roughness Ra (μm) |
|---|---|---|---|---|---|---|---|
| AL-CLSTM | Depth: 18μm. Speed: 42m/s. Feed: 2500mm/min | 2000 | 0.01 | 0.04532 | 0.05578 | 310 | 0.1985 |
| | Depth: 19μm. Speed: 44m/s. Feed: 3000mm/min | 2000 | 0.02 | 0.03857 | 0.04863 | 290 | 0.1857 |
| | Depth: 17μm. Speed: 43m/s. Feed: 3500mm/min | 2500 | 0.015 | 0.04289 | 0.05134 | 330 | 0.1923 |
| | Depth: 18.5μm. Speed: 43.5m/s. Feed: 3200mm/min | 3000 | 0.012 | 0.03210 | 0.04125 | 400 | 0.1902 |
| | Depth: 17.5μm. Speed: 42.5m/s. Feed: 3100mm/min | 1500 | 0.008 | 0.05001 | 0.06012 | 270 | 0.2050 |
| | Depth: 18μm. Speed: 43m/s. Feed: 3000mm/min | 2000 | 0.04 | 0.03746 | 0.04621 | 295 | 0.1980 |

to 0.1624 µ m. This change can be explained by the stress distribution and heat generation mechanism during the grinding process. A larger grinding depth increases the cutting area, allowing the cutting force and friction force during the cutting process to be fully utilized, which helps improve the efficiency of material removal and reduce the local heat generated during the grinding process, thereby reducing microcracks caused by thermal stress. Meanwhile, the increase in grinding speed (from 40m/s to 45m/s) also had a positive impact on the prediction accuracy of the model. During the grinding process, a higher linear velocity of the grinding wheel can reduce the contact time between the material and the grinding wheel, decrease the accumulation of frictional heat, and mitigate its adverse effects on surface quality. Meanwhile, the increase in grinding speed (from 40m/s to 45m/s) also had a positive impact on the prediction accuracy of the model. During the grinding process, a higher linear velocity of the grinding wheel can reduce the contact time between the material and the grinding wheel, decrease the accumulation of frictional heat, and mitigate its adverse effects on surface quality.

The linear speed of the grinding wheel, grinding depth, and feed rate are key factors affecting surface roughness. The linear velocity of the grinding wheel affects the contact frequency and cutting force between the abrasive particles and the workpiece, which in turn affects the grinding efficiency and temperature, while the grinding depth and feed rate directly affect the material removal rate and grinding force. The selection of these parameters is to explore their impact on surface roughness, in order to find a balance between improving grinding efficiency and ensuring quality.

Different parameters of ceramic grinding processing can have an impact on surface roughness. Firstly, the selection range for the linear velocity of the grinding wheel (unit: m/s) is 30–45 m/s. The grinding line speed has a significant impact on surface roughness, and higher line speeds usually reduce cutting forces, thereby reducing surface roughness. Secondly, the setting range of grinding depth (unit: mm) varies from 0.005 to 0.015 mm. The selection of this parameter is based on the characteristics of ceramic materials, and excessive grinding depth may lead to increased tool wear and surface damage. Finally, the selection range of feed rate (unit: mm/min) is 1000–5000 mm/min. The feed rate is directly related to the material removal rate and surface quality. A reasonable feed rate can improve production efficiency while ensuring the fineness of the processed surface. The experimental results are shown in Table 4.

Table 4 summarizes the experimental results of the influence of different grinding parameters on the surface roughness of ceramic grinding. With the increase of grinding wheel linear velocity and moderate adjustment of grinding depth, the AL-CLSTM model exhibits lower surface roughness Ra values under various experimental conditions. For example,

Table 4. Experimental results of the influence of different parameters on surface roughness in ceramic grinding processing.

| Experiment number | Wheel linear speed (m/s) | Grinding depth (mm) | Feed rate (mm/min) | Surface roughness Ra (µm) | Reverse grinding Ra (µm) | Relative error (%) |
|---|---|---|---|---|---|---|
| 1 | 30 | 0.008 | 1000 | 0.2012 | 0.2228 | 2.78 |
| 2 | 35 | 0.010 | 2000 | 0.1964 | 0.2157 | 1.83 |
| 3 | 40 | 0.012 | 3000 | 0.1908 | 0.2089 | 1.37 |
| 4 | 45 | 0.015 | 4000 | 0.1876 | 0.2063 | 0.95 |
| 5 | 32 | 0.005 | 1500 | 0.1943 | 0.2246 | 2.12 |
| 6 | 38 | 0.011 | 2500 | 0.1917 | 0.2118 | 1.74 |
| 7 | 42 | 0.009 | 3500 | 0.1849 | 0.2034 | 1.05 |
| 8 | 34 | 0.014 | 5000 | 0.1894 | 0.2121 | 1.13 |

when the grinding wheel linear velocity is set to 45 m/s and the grinding depth is 0.015 mm, the Ra value drops to 0.1876 μm, indicating that the optimization of grinding parameters significantly improves the machining quality. Grinding force, heat generation, and material properties are the main physical factors that affect the results. Moderate grinding depth can reduce cutting force and tool damage, while high linear velocity effectively reduces cutting time, reduces heat accumulation, and avoids cracks in ceramic materials caused by overheating during processing. In addition, the high hardness and brittleness of ceramic materials require stress control during grinding to prevent crack propagation. Therefore, the reasonable selection of grinding wheel line speed and feed rate is the key to improving surface smoothness. The experimental results of the influence of grinding wheel material particle size, lubricant composition, and processing time on the surface roughness of ceramic bearings are shown in Table 5.

The experimental results in Table 5 indicate that as the grain size of the grinding wheel increases, the surface roughness usually shows a decreasing trend. When using a 100 μm CBN grinding wheel and water-based lubricant, the surface roughness reaches 0.2104 μm, while when using a 150 μm CBN grinding wheel, the surface roughness decreases to 0.1987 μm. Although in the experiment with a grinding time of 15 minutes, the application of oil-based lubricant achieved better results of 0.1873 μm at a particle size of 200 μm, indicating a significant impact of lubricant composition on the grinding effect. Overall, during the grinding process, the smaller the particle size of the grinding wheel, combined with appropriate lubricant composition and processing time, can more effectively reduce surface roughness and improve the overall processing quality of ceramic bearings. The final ceramic bearing obtained is shown in Fig 11.

Fig 11 shows six ceramic bearings treated with different grinding parameters, as well as five precisely molded ceramic bearings. These products have been systematically studied, including careful experiments on the effect of each grinding parameter on SR. High-performance ceramic materials such as zirconia and silicon nitride are selected for the GP. The grinding quality is accurately predicted by means of an AL-CLSTM. The surface finish of these ceramic bearings is evident, with SR Ra values reaching industrial standards and demonstrating superior machining quality. In particular, it should be noted that the surface smoothness is improved after adjusting the grinding parameters, while the prediction model of the grinding quality successfully guided the optimization process. In the experiments, an active feedback mechanism is employed to realize intelligent control of the GP by monitoring the grinding status in real time. Meanwhile, these ceramic bearings exhibit excellent anti-wear performance and stability after series grinding and treatment for demanding industrial applications.

Table 5. Impact of Different Parameters on Ceramic Bearing Surface Roughness.

| Experiment No. | Wheel Material | Grain Size (μm) | Lubricating Fluid Composition | Grinding Time (min) | Surface Roughness Ra (μm) | Relative Error (%) |
|---|---|---|---|---|---|---|
| 1 | CBN | 100 | Water-based | 15 | 0.2104 | 1.85 |
| 2 | CBN | 150 | Water-based | 15 | 0.1987 | 1.72 |
| 3 | CBN | 200 | Oil-based | 20 | 0.1873 | 1.66 |
| 4 | CBN | 300 | Oil-based | 20 | 0.1861 | 1.50 |
| 5 | Diamond | 100 | Water-based | 10 | 0.2022 | 1.80 |
| 6 | Diamond | 150 | Oil-based | 25 | 0.1829 | 1.40 |
| 7 | Diamond | 200 | Oil-based | 25 | 0.1725 | 1.20 |
| 8 | CBN | 250 | Water-based | 30 | 0.1974 | 1.80 |

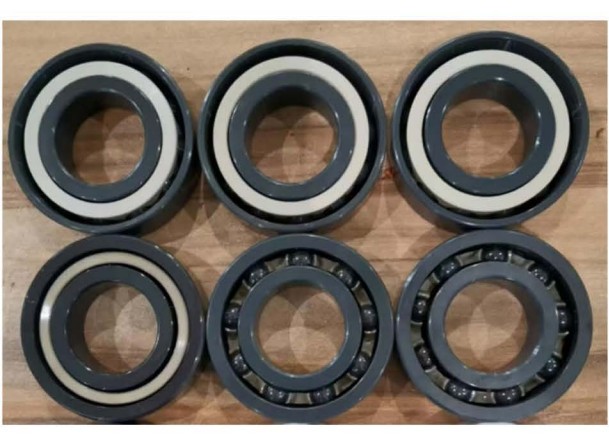
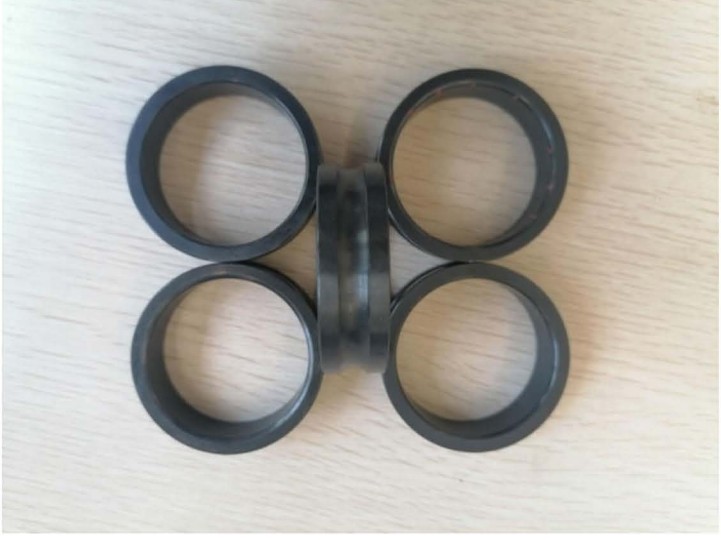

**Fig 11. Ceramic bearings.**

## 4. Discussion

In the grinding process of ceramic bearings, the influence of different grinding parameters on surface roughness is closely related to multiple factors, including grinding speed, grinding depth, and feed rate[32]. These factors are considered as key variables affecting the material removal mechanism in tribology theory[33]. The experimental results show that as the grinding speed increases, the surface roughness significantly decreases, indicating that optimizing the grinding speed plays a key role in improving the surface smoothness of ceramic materials, which is consistent with the theory of grinding force and contact state in tribology. Meanwhile, the combination of grinding depth and feed rate also has a significant impact on the generation of grinding force and heat. According to Hertz contact theory, as the grinding depth increases, the contact surface pressure also increases, which not only leads to the generation of more heat but also causes the failure of the grinding tool[34]. Therefore, appropriate selection of grinding depth can ensure machining quality and reduce tool wear. The setting of feed rate is directly related to the cutting quantity. For brittle ceramic materials, a lower feed rate helps to reduce instantaneous thermal shock and lower the risk of cracking during the machining process.

Comparing the experimental results of the study with relevant literature can not only provide a deeper understanding of the impact of grinding parameters on surface quality but also highlight the unique advantages of this study. For example, S. M. Lee's research suggests that the mechanism by which different abrasives affect surface roughness is closely related to their cutting performance and the dissipation of grinding heat. This indicates that under the same grinding conditions, the characteristics of the material determine the differences in machining performance. In this study, a step further is taken by precisely quantifying the surface roughness of ceramic materials within a specific range, fluctuating between 1.03% and 2.86%. This detailed quantification allows for a more accurate assessment of the material's behavior during grinding compared to previous general descriptions in the literature, and it is consistent with the description of the interaction between material removal and grinding in tribology. Meanwhile, the new grinding head developed by L. Jiang et al. significantly reduced surface roughness and grinding force, demonstrating the potential for improving tool design to achieve

higher grinding quality. While that innovation focused on the tool design aspect, this study adopts a more comprehensive approach. Not only the impact of tool design is considered, but also a variety of grinding parameters and their interactions are systematically explored. Through a series of carefully designed experiments, the optimal combination of these parameters that maximizes the improvement in machining efficiency is identified, which is consistent with the goal of optimizing grinding parameters to improve machining efficiency in this study and provides a more holistic solution for achieving high grinding quality.

In the comparison of different grinding methods, X. Wang's surface finishing technology for precision bearing rolling elements emphasizes the balance between surface roughness and geometric errors during the grinding process. In contrast, this study not only supports the view that improving surface quality requires maintaining cost - effectiveness and production efficiency but also presents a novel methodology for evaluating and optimizing these aspects simultaneously. A unique evaluation model is developed that takes into account multiple factors specific to the experimental setup and application scenarios, enabling more targeted and practical recommendations for industrial applications.

Overall, by comparing with existing literature, it becomes evident that this study offers distinct contributions and advantages in furthering the understanding and improvement of grinding processes and surface quality.

When conducting a systematic analysis of the uncertainty of the obtained results, it is considered that the uncertainty mainly comes from the accuracy of the measuring instruments, environmental changes, and differences in the characteristics of the materials themselves. To quantify these uncertainties, multiple repeated experiments were conducted and the standard deviation (SD) was calculated for each set of experimental data. Finally, the uncertainty of the surface roughness results evaluated using the uncertainty propagation formula was $\pm$ 0.003 $\mu$ m. This analysis provides an important basis for evaluating the reliability of experimental data.

Although this study has drawn meaningful conclusions, data acquisition may be limited by factors such as sample size, degree of control over the experimental environment, and variables that may not have been fully considered during the grinding process, including tool wear status, material changes during the grinding process, fluctuations in grinding force due to machine vibration, and the influence of coolant properties on the grinding interface. These limitations may affect the accuracy of experimental results. So in future studies, the sample size and number of experimental replicates should be increased, and the impact of various factors, such as those mentioned above, on the results should be further explored to improve the comprehensiveness and credibility of the research.

## 5. Conclusion

Traditional grinding quality prediction methods suffer from insufficient accuracy and poor real-time performance. Therefore, the study addressed the quality prediction problem in ceramic bearing GP, aiming to improve the control and PA of SR in the GP. The study proposed an AL-CLSTM model combined with deep learning techniques to optimize the grinding parameters and improve the quality PA. The study verified the effectiveness of model training through several experiments. Systematic tests were conducted under different grinding conditions, including parameters such as grinding WLS, GD and feed rate. The research results show that the AL-CLSTM model exhibits excellent predictive performance after 2000 iterations, with an average training loss of 0.03572, an average prediction error of 0.04589, and a surface roughness Ra value of 0.1985 $\mu$ m. Under different grinding parameter settings, the average training loss and prediction error of the AL-CLSTM model are relatively low. For example, when the grinding depth is 20 $\mu$ m, the average training loss is 0.04237 and the average prediction error is 0.05312. Although the study observed an

improvement in surface roughness, the variation in roughness was only 0.001 to 0.002 μ m due to measurement errors (e.g., ± 0.003 μ m). This means that the observed improvement may not exceed the range of measurement error. Therefore, in further research, it is necessary to optimize the measurement process to reduce experimental errors and ensure accurate reflection of the impact of grinding parameter optimization on surface quality. Meanwhile, based on statistical analysis, as the sample size increases, it is expected to obtain more significant results to verify the true impact of grinding operations on the surface quality of ceramic bearings.

## Supporting information

**S1 Data. Captions for the Supporting Information: Minimal Data Set.**
(DOCX)

## Author contributions

**Conceptualization:** Yuhou Wu.

**Data curation:** Longfei Gao.

**Formal analysis:** Junxing Tian.

**Funding acquisition:** Yuhou Wu.

**Investigation:** Jian Sun.

**Methodology:** Longfei Gao.

**Project administration:** Jian Sun.

**Software:** Yuhou Wu.

**Supervision:** Longfei Gao, Junxing Tian.

**Visualization:** Junxing Tian.

**Writing – original draft:** Jian Sun.

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
