## [Decision Letter · Decision Letter 0]

22 Dec 2024

PONE-D-24-55429Active Learning Regression Quality Prediction Model and Grinding Mechanism for Ceramic Bearing Grinding ProcessingPLOS ONE

Dear Dr. Gao,

Thank you for submitting your manuscript to PLOS ONE. After careful consideration, we feel that it has merit but does not fully meet PLOS ONE’s publication criteria as it currently stands. Therefore, we invite you to submit a revised version of the manuscript that addresses the points raised during the review process.

**ACADEMIC EDITOR: **As per the reviewer's feedback, I am recommending "Major Revision" for this paper. Authors must go through each of the reviewer comments very seriously and make a sincere point-to-point response and improve their paper accordingly. If authors will be able to undertake this task then I will be happy to reconsider my decision. 

We look forward to receiving your revised manuscript.

Kind regards,

Himadri Majumder, Ph.D

Academic Editor

PLOS ONE

3. Thank you for stating the following financial disclosure:  [The research is supported by National Natural Science Foundation of China in 2024: Research on Dynamic Characteristics Control of Full Ceramic Ball Bearing Retainers for Wide Temperature Range Oil free Lubrication Conditions (Fund No. 52405123); National Natural Science Foundation of China in 2021: Research on the Characteristics and Discrimination Mechanism of Lubricating Oil Film in Multi Field Coupled All Ceramic Ball Bearings Based on Elastic Flow Pressure Lubrication (Fund No. 52105196).].  Please state what role the funders took in the study.  If the funders had no role, please state: "The funders had no role in study design, data collection and analysis, decision to publish, or preparation of the manuscript." If this statement is not correct you must amend it as needed. Please include this amended Role of Funder statement in your cover letter; we will change the online submission form on your behalf.

Additional Editor Comments:

As per the reviewer's feedback, I am recommending "Major Revision" for this paper. Authors must go through each of the reviewer comments very seriously and make a sincere point-to-point response and improve their paper accordingly. If authors will be able to undertake this task then I will be happy to reconsider my decision.

Reviewers' comments:

Reviewer's Responses to Questions

**Comments to the Author**

1. Is the manuscript technically sound, and do the data support the conclusions?

Reviewer #1: Yes

Reviewer #2: Partly

Reviewer #3: Partly

2. Has the statistical analysis been performed appropriately and rigorously? 

Reviewer #1: N/A

Reviewer #2: Yes

Reviewer #3: N/A

3. Have the authors made all data underlying the findings in their manuscript fully available?

Reviewer #1: Yes

Reviewer #2: Yes

Reviewer #3: Yes

4. Is the manuscript presented in an intelligible fashion and written in standard English?

Reviewer #1: Yes

Reviewer #2: No

Reviewer #3: Yes

5. Review Comments to the Author

Reviewer #1: 1. At the end of the Introduction section, write the scientific hypotheses.

2. How are the parameters shown in table 1 selected?

3. How the selected grinding parameters are shown in table 4. (speed, depth and feed)? Why are they representative for your research?

4. What is the uncertainty of the obtained results?

5. In general, the results discussion must be enlarged and more comparisons with literature data should be reported. The authors should go more in depth with the interpretation of the results;

6. Please write more theoretical support in the discussion section, so far only limited to stating the results. You need to extract valuable ideas from the data.

Reviewer #2: The reviewer suggests that the manuscript can be accepted after the author resolves the conflicting parts, clarifies the ambiguous parts and corrects the formatting issues.

1. On page 9, in lines 4 and 15, “cBN” should be changed to “CBN”.

2. On page 10, in the ninth line from the bottom, the sequence of “cutting, plowing, and friction” should be changed to “scraping, plowing, cutting”.

3. On page 11, Figure 1 does not effectively demonstrate the three stages of abrasive cutting, and Figure 2 should be centered.

4. On page 12, both Formula 1 and Formula 2 represent the calculation of the critical values for the generation of radial cracks, and further elaboration is needed.

5. On page 19, the conclusion in the seventh line differs from what is depicted in Figure 8(b). In the tenth line, "a range of 0.10 to 0.10" should be corrected as it represents a single value rather than a range, and this should be revised in conjunction with Figure 8(c). Additionally, the legend in Figure 9(c) does not correspond with the colors in the image.

6. It is recommended that each value in the "Average training loss" column should be written on a separate line instead of spanning two lines.

7. On page 23, the two graphs in Figure 11 are of different sizes and need to be adjusted to match. This suggests that for consistency and clarity, the figures should be resized so that they are of equal dimensions within the figure.

8. In the conclusion section of the article, examples are needed to illustrate the situations in which the accessibility of data may be restricted.

Reviewer #3: The manuscript focuses on predicting the quality of ceramic bearings during grinding. The manuscript presents an active learning regression model for the construction and optimization of a roughness model. The variable parameters are linear grinding wheel speed, grinding depth and feed rate. The conducted experiments showed that the mentioned parameters affect the surface roughness of the workpiece.

The presented experimental data are useful for practical applications. However, the scientific value of the regression model needs to be adjusted. For this purpose, it is necessary to answer the following principal comments.

1. The influence of grinding modes on surface roughness is known. The authors have not found any differences for grinding ceramic bearings.

2. There is no physical explanation for the results obtained.

3. The results for roughness reduction are close to the measurement error.

4. When grinding ceramic bearings, significantly lower roughness values than Ra = 0.2 µm are achievable.

5. It is not sufficient to evaluate the quality of machining by one roughness parameter alone.

6. The surface roughness during grinding is more influenced by the grain size of the grinding wheel material, the composition of the lubricating fluid, and the machining time. These factors were not considered in the model.

7. Data on grinding machine, grinding wheel, measuring devices are not given.

6. PLOS authors have the option to publish the peer review history of their article (what does this mean? ). If published, this will include your full peer review and any attached files.

**Do you want your identity to be public for this peer review?** For information about this choice, including consent withdrawal, please see our Privacy Policy .

Reviewer #1: No

Reviewer #2: No

Reviewer #3: No

---

## [Author Response · Author response to Decision Letter 0]

7 Feb 2025

The manuscript has been modified according to comments.

---

## [Decision Letter · Decision Letter 1]

20 Feb 2025

Active Learning Regression Quality Prediction Model and Grinding Mechanism for Ceramic Bearing Grinding Processing

PONE-D-24-55429R1

Dear Dr. Gao,

We’re pleased to inform you that your manuscript has been judged scientifically suitable for publication and will be formally accepted for publication once it meets all outstanding technical requirements.

Kind regards,

Himadri Majumder, Ph.D

Academic Editor

PLOS ONE

Additional Editor Comments (optional):

Reviewers' comments:

Reviewer's Responses to Questions

**Comments to the Author**

1. If the authors have adequately addressed your comments raised in a previous round of review and you feel that this manuscript is now acceptable for publication, you may indicate that here to bypass the “Comments to the Author” section, enter your conflict of interest statement in the “Confidential to Editor” section, and submit your "Accept" recommendation.

Reviewer #1: All comments have been addressed

Reviewer #2: (No Response)

Reviewer #3: (No Response)

2. Is the manuscript technically sound, and do the data support the conclusions?

Reviewer #1: Yes

Reviewer #2: (No Response)

Reviewer #3: Yes

3. Has the statistical analysis been performed appropriately and rigorously? 

Reviewer #1: N/A

Reviewer #2: (No Response)

Reviewer #3: Yes

4. Have the authors made all data underlying the findings in their manuscript fully available?

Reviewer #1: Yes

Reviewer #2: (No Response)

Reviewer #3: Yes

5. Is the manuscript presented in an intelligible fashion and written in standard English?

Reviewer #1: Yes

Reviewer #2: (No Response)

Reviewer #3: Yes

6. Review Comments to the Author

Reviewer #1: (No Response)

Reviewer #2: In this manuscript, the authors delve into quality prediction in ceramic bearing grinding, emphasizing the influence of grinding parameters on surface roughness. They employ an active learning regression model for model construction and optimization, and conduct empirical analysis of surface quality under different grinding conditions. Additionally, various deep learning models are utilized for quality prediction experiments, covering a range of grinding parameters such as grinding wheel linear speed, grinding depth, and feed rate to ensure model accuracy and reliability. The experimental results show that increasing the grinding depth to 21 μm significantly reduces the model's average training loss to 0.03622 and the surface roughness Ra value to 0.1624 μm. The study also finds that increasing the grinding wheel linear velocity and moderately adjusting the grinding depth can significantly improve machining quality, with the Ra value dropping to 0.1876 μm at a linear velocity of 45 m/s and a grinding depth of 0.015 mm. These findings not only provide theoretical support for ceramic bearing grinding but also offer a basis for optimizing grinding parameters in actual production, holding significant industrial application value. Therefore, I suggest it be accepted.

Reviewer #3: Comments to the Author

1. If the authors have adequately addressed your comments raised in a previous round of review and you feel that this manuscript is now acceptable for publication, you may indicate that here to bypass the “Comments to the Author” section, enter your conflict of interest statement in the “Confidential to Editor” section, and submit your "Accept" recommendation.

7. PLOS authors have the option to publish the peer review history of their article (what does this mean? ). If published, this will include your full peer review and any attached files.

**Do you want your identity to be public for this peer review?** For information about this choice, including consent withdrawal, please see our Privacy Policy .

Reviewer #1: No

Reviewer #2: No

Reviewer #3: No

---

## [Editor Report · Acceptance letter]

PONE-D-24-55429R1

PLOS ONE

Dear Dr. Gao,

I'm pleased to inform you that your manuscript has been deemed suitable for publication in PLOS ONE. Congratulations! Your manuscript is now being handed over to our production team.

Kind regards,

on behalf of

Dr. Himadri Majumder

Academic Editor

PLOS ONE